# Retaining public health volunteers beyond COVID-19

**Ameeta Retzer**[1,2], **Janet Jones**[1], **Sarah Damery**[1], **Habib Ullah**[3], **Modupe Omonijo**[3], **Justin Varney**[1,3], **Kate Jolly**[1]*

**1** Institute of Applied Health Research, University of Birmingham, Birmingham, United Kingdom, **2** Centre for Patient Reported Outcomes Research, Institute of Applied Health Research, University of Birmingham, Birmingham, United Kingdom, **3** Birmingham City Council, Birmingham, United Kingdom

* c.b.jolly@bham.ac.uk

**Data Availability Statement:** All relevant data are within the paper and its Supporting Information files.

**Funding:** This study was funded by the National Institute for Health Research (NIHR) Applied Health

## Abstract

### Objectives

The COVID-19 pandemic has led to a change in people's volunteering behaviours; participation has increased in informal volunteering (giving unpaid help to those who are not a relative) while decreasing in formal volunteering (unpaid help to groups or clubs). There is an interest from stakeholders who have experienced increased participation in maintaining the positive patterns of volunteering, aligning with National Health Service (NHS) objectives and realising benefits in a wider public health context. This research uses a local COVID-19 public health volunteering programme case study to explore the volunteer's journey and perspective using volunteers' reported experiences to consider the potential for volunteer retention and role expansion into other public health issues beyond the COVID-19 pandemic.

### Methods

Recruitment was undertaken by Birmingham City Council Public Health Team via the COVID-19 Community Champions programme mailing list. Semi-structured focus group discussions, one-to-one interviews and email interviews were conducted with volunteers. Data were analysed through directed thematic analysis using an iteratively developed coding frame.

### Results

Data were collected from three focus group discussions, four interviews, and one email interview involving a total of 16 participants. Six themes were identified: volunteer motivations and expectations; volunteer management; programme organisation; feeling valued; continued need for role, and interest in new responsibilities.

### Conclusion

Our findings indicate that the factors which are conducive to volunteer recruitment, retention and re-purposing were: maintaining the original terms of engaging with the volunteering

Research Collaboration West Midlands. The funders had no role in study design, data collection and analysis, decision to publish, or preparation of the manuscript.

**Competing interests:** The authors have declared that no competing interests exist.

opportunity (including retaining the original brief and remit), adjusting these through consultative processes with an emphasis on seeking permission from the volunteers already involved and ensuring a reliable and consistent management and support structure. While some of the learning is specific to the local volunteer programme in question and the context of the COVID-19 pandemic, there are lessons that can be generalised to other scenarios and settings.

## Introduction

The COVID-19 pandemic has led to a change in people's volunteering behaviours. In 2020–2021 in the UK, formal volunteering (unpaid help to groups or clubs, for example befriending or mentoring people) fell to its lowest at 14 million people in England since 2013–2014 after an increase in 2019–2020, though more people (25 million people in England) have been involved in informal volunteering (giving unpaid help to individuals who are not a relative) than ever before [1]. Some segments of the volunteering sector have faced unprecedented financial challenges [2] and are increasing their paid workforce [3], while others, such as the National Health Service (NHS) Volunteer Responder scheme launched in response to the pandemic, far exceeded their volunteering targets [4]. Amidst this variation, there is an interest from stakeholders that have experienced increased participation in maintaining the positive patterns of volunteering from the pandemic. This aligns with the UK NHS long-term plan which includes an aim to double the number of volunteers across the organisation as part of its move towards a 'participation culture' [5]. The personal benefits of volunteering are becoming better understood [6]. For example, people may volunteer to feel connected to others, and may find it emotionally cathartic and comforting to work with others towards a shared goal [7]. Furthermore, the use of volunteers in a public health context may allow for creative, autonomous programmes that are rooted in communities while permitting individuals to work flexibly [8].

The most effective way to harness the wave of enthusiasm for volunteering and maintain it beyond the pandemic is yet to be established [9]. Spontaneous volunteering, where volunteers emerge in response to a sudden need caused by a crisis in an unplanned and ad hoc mode [10], has been observed in a range of cases including natural disasters [11–13] and specific historical events [14]. Other events that have inspired ad hoc periods of volunteering include sporting events such as the Olympics or Commonwealth Games. These have led to similar post-event reflective exercises to understand and sustain the momentum around volunteering [15, 16]. There are a number of challenges inherent to successful volunteering programmes. These include ensuring there is a strategic approach to how volunteering will contribute to organisational aims; carefully considering skillsets so volunteers take up appropriate roles while ensuring quality; managing and supporting volunteers, particularly in relation to paid staff [17].

Gaskin's model of volunteer involvement consists of four stages, starting with the non-volunteer (the doubter) and progresses through the starter and the doer to the long-term volunteer (the stayer), and details how organisations can facilitate and promote volunteer longevity [18]. The aim would be for a volunteering programme, once aware of these stages, to assist in the transition from one stage to the next. Gaskin identifies eight potential barriers to this progression and details how these may be addressed by organisations. Gaskin's model of volunteer role progression, pressure points, and suggestions for effective actions informed the data collection and interpretation. Further research is needed to understand the motives for volunteering and how these inform strategies for volunteer retention [19].

Among the range of volunteer programmes launched in response to the COVID-19 pandemic is the COVID-19 Community Champions programme which was established throughout England by local councils to work directly with their local populations. In Birmingham, to ensure residents are up to date with the latest guidance and advice, Birmingham City Council's Public Health Team shares information using a range of formats including webinars and emails for Champions to disseminate through their personal, professional, and community networks [20]. The only eligibility criteria for joining is that the individuals must be aged 18 or over, reside in Birmingham, and be able to receive emails. During the course of the pandemic, several hundred people have become COVID-19 Community Champions via Birmingham City Council Public Health Team. Types of information shared are described in Box 1.

> ### Box 1. Information shared by local authority with the COVID champions
>
> Information distributed via the volunteers included advice relating to:
>
> - Accessing personal protective equipment;
>
> - Wearing face coverings in public places;
>
> - Local COVID-19 infection and hospitalisation rates;
>
> - Self-isolation rules;
>
> - Government stay at home guidance;
>
> - Access to financial support when self-isolating;
>
> - Access to support when shielding, and
>
> - Locating testing and vaccination centres.

Birmingham has a highly diverse population with high levels of socio-economic deprivation; the population experiences high levels of inequalities in health outcomes with variation across ethnic groups; and an above average proportion of its population living in overcrowded conditions [21]. Within this context of high need for interventions to improve the public health of residents, there is potential to use volunteers to engage with other public health issues either by expanding the existing Community Champion role or by learning from the experience to recruit new volunteers. The local authority was keen to explore how volunteers engaged with the COVID-19 Community Champions programme would feel about being asked to address other public health issues as the need for local COVID information declines.

This research uses the Birmingham COVID-19 Community Champions programme as a case study to explore the experiences of volunteers and consider volunteer retention in this context, alongside the extent to which there is potential for the role to be maintained beyond the COVID-19 pandemic and revised to include other public health issues.

## Methods

This qualitative research is reported in accordance with COREQ [22] (S1 Table) and GRIPP2-SF [23] reporting guidelines.

## Patient and public involvement

The National Institute for Health Research (NIHR) Applied Research Collaboration West Midlands (ARCWM) Public Health Theme Patient and Public Involvement (PPI) group contributed to the design of this study. The group met regularly and the researchers presented specific aspects of the research (e.g. remote focus group facilitation, setting expectations during the consenting process), requesting their feedback and advice. The PPI group met on three occasions to advise on study conduct, relating to preparing participants and clarifying their expectations during the advance consenting session, remote focus group discussion facilitation, and sharing study findings.

## Recruitment

Participants for the study were adult volunteers (aged 18 and above) on the COVID-19 Community Champions programme. Recruitment was undertaken via the COVID-19 Community Champions programme, invitations to participate in the study were shared by Birmingham City Council Public Health Team with the individuals on the programme's mailing list. The study was discussed during a routinely scheduled COVID-19 Community Champion webinar organised by Birmingham City Council Public Health Team, attended by volunteers and members of the Public Health team. Recruitment continued on a rolling basis until data saturation was reached [24]. Potential participants were provided with the researcher's (AR) contact information and upon making contact, were sent a Participant Information Sheet (PIS) and consent form, and those interested in participating were invited to take part in a focus group discussion. A pragmatic approach was taken and where necessary, one-to-one interviews via zoom or telephone, or email interviews were offered at a range of times to accommodate the schedules of participants unable to attend a focus group. Verbal consent was taken in advance. Consent was taken after the potential participant made contact with the researcher, reviewed the PIS and consent form in their own time, had their questions answered to their satisfaction, and a convenient time-slot had been identified. Phone or video conferencing software was used. The researcher talked the interviewee through each statement on the consent form, and the participant was asked to give their agreement to each statement. This process was audio-recorded and the consent recording was saved among the study records as proof of consent. The consent form completed by the research fellow was sent to the participant for their own records. Written consent was not used in this study as verbal consent enabled participation from a wider pool of people as it did not require postage and/or access to printing and scanning equipment, and also minimised contact in light of the ongoing COVID-19 pandemic.

## Data collection

Due to the evolving COVID-19 situation, this research was undertaken rapidly so the findings could be relayed and implemented before they became outdated [25]. Data collection took place between July and September 2021. Participants were not known to researchers prior to study commencement and participants' knowledge of the researcher related only to their involvement in the study. A preliminary topic guide was formulated in advance, informed by the research aims. The topic guide was iteratively refined to explore emerging themes and broadly covered: participants' experience within their role; perceived strengths and weaknesses of the programme in its current form; their expectations of the programme; views about how the programme might be improved; activities undertaken; interest in an expanded role, and whether they would recommend the programme to others. The same topic guide was used for the focus groups, one-to-one interviews and email interviews (S1 Appendix). Data collection

was undertaken by two female health researchers with doctoral and post-doctoral experience of qualitative methodology. One researcher facilitated sessions (AR) and the other (JJ) made detailed field-notes. Sessions were conducted using video-conferencing software and were audio-recorded. Verbatim transcripts were automatically generated by the video-conferencing software for reference during the analysis. After each interview, the recording was replayed by AR and the field-notes were supplemented as required.

## Analysis

In the interests of rapidly relaying findings, field-notes and recordings rather than the transcripts were analysed. Directed thematic analysis [26] was undertaken by one researcher (AR). An initial framework was developed informed by the research aims and Gaskin's model of volunteer involvement [18]. The field-notes were read and a flexible and iterative approach was used to continually develop and refine the coding frame, allowing for the emergence of novel themes. Additional codes were developed and integrated as analysis progressed and the framework was modified as required [27]. The coding frame and sample codes were checked by the other qualitative researcher (JJ). Disagreements were resolved through discussion. Illustrative quotes were identified from the field-notes and drawn from the audio-recordings or automatically generated transcripts.

## Results

A total of 16 Community Champions participated in the study, spread across three focus group discussions (n = 11), four one-to-one interviews, and one email interview. Seven individuals expressed interest in the study but did not participate. Focus group discussions lasted between 1 and 2 hours; interviews lasted approximately 40 minutes. Participants were drawn from across the city, from 16 different wards, representing broad demographic characteristics (Table 1).

Participants reported a wide range of joining dates covering the breadth of the pandemic. Participants described a wide variety of activities that they undertook in their role. Participants reported sharing information with their personal, local, and professional networks. Some would select specific items of interest, simplify the text to increase accessibility, or translate into a range of languages depending on their audience. Some would forward entire emails to their networks, unedited. Participants used newsletters, group chats, emails, and social media; would share information face to face with their network and in key community spaces; print leaflets and make copies, handing these out, putting through letterboxes, or leaving in visible parts of the community; check in via telephone with people to ensure they had the correct information.

Six themes were identified: volunteer motivations and expectations; volunteer management; programme organisation; feeling valued; continued need for role, and interest in new responsibilities.

## Volunteer motivations and expectations

Participants joined the programme for several reasons including personal situations such as previous experience of volunteering or their current employment status (several were retired and one reported seeking employment):

*I am the neighbourhood watch coordinator for my local community here and basically Birmingham City Council reached out into that network (FGD A)*

*Because I am unemployed as well as looking for work. . . I am keeping my skills going by being actively involved in the community (Interview2)*

**Table 1. Study participant characteristics.**

| Characteristic | n | Group A* | Group B | Group C | Interviews |
|---|---|---|---|---|---|
| AGE | | | | | |
| 31–40 | 2 | | 2 | | |
| 41–50 | 2 | 1 | | | 1 |
| 51–60 | 4 | | 2 | | 2 |
| 61–70 | 8 | 3 | 1 | 2 | 2 |
| GENDER | | | | | |
| Male | 7 | 1 | 4 | 1 | 2 |
| Female | 9 | 3 | 1 | 1 | 3 |
| ETHNICITY | | | | | |
| White British | 10 | 3 | 2 | 1 | 4 |
| White Irish | 1 | | | 1 | |
| White other | 1 | | 1 | | |
| Asian Bangladeshi | 1 | 1 | | | |
| Asian British Indian | 1 | | | | 1 |
| African | 1 | | 1 | | |
| Caribbean | 1 | | 1 | | |
| EMPLOYMENT STATUS | | | | | |
| Retired | 5 | 2 | 1 | 1 | 1 |
| In employment | 10 | 2 | 4 | 1 | 3 |
| Seeking employment | 1 | | | | 1 |

*These are referred to in the identifiers as Focus Group Discussion (FGD) A, B, C etc.

*My son is a teenager and he was hearing all sorts of things through social media and so on, so there seems to be so many mixed messages (FGD A)*

Others had reasons specifically relating to COVID-19, such as special access to trusted information that would otherwise be unavailable or difficult to source, personal experience of COVID, altruism, or it being complementary to their professional roles or their work with specific communities:

*My main role was working for BAME communities... Asian people are more vulnerable and dying and having COVID, and that was creating like kind of a fear in Community and then to resolve that, then vaccine hesitancy... (FGD A)*

Some participants reported that the role was clearly described as specifically related to the dissemination of COVID-19 information and that they would regularly receive up to date, locally relevant content to facilitate this. Others described having unclear expectations of what the programme would entail and some reservations about becoming a volunteer. These included being wary of the responsibility, time commitment, and sole use of online resources and interactions. Several reported that once registered, the workload and brief became clearer and were reassured it was a manageable commitment:

*I was worried about the time commitment... but I mean it soon became very clear that that wasn't an issue at all, (FGD A)*

Participants differed in their perceptions about the support that they expected to receive from the programme organisers in the fulfilment of their role. Several stated that they

understood the programme organisers were likely to be a small team, working remotely, and with limited capacity. Some reported that support provision for volunteers was not mentioned upon registration as a volunteer, so they had no expectations in relation to this.

## Volunteer management

Management needs and expectations varied between the participants. Several participants reported never having made contact with the programme organisers outside of routine emails and regular (fortnightly) webinars, and felt that the webinar Question and Answer sessions were sufficient:

> *I like the fortnightly zoom meetings because you're putting them on the spot, and they are very good at going through the questions that are asked in the meeting either verbally or in the chat (FGD A)*

> *I raised that the messaging wasn't getting out there about lateral flow tests and we were having a problem with people saying I'm not doing them because they're not reliable and it was really helpful to be able to say to the director of public health and his team. . . (FGD A)*

For these participants, the interval between webinars was seen as appropriate and any questions they might have were general enough that others would be likely to raise them if they were unable to attend or ask themselves. For these participants, the availability of recorded webinars was especially important:

> *I can watch them after, you know I'm willing to go for the time that the majority of people can do. (Interview3)*

These individuals were satisfied with their interactions and did not perceive there to be any further need.

For some, the perceived speed and quality of the programme organisers' response to emails was closely linked to their overall satisfaction with volunteer management. A number of participants reported making contact with specific requests or observations, and that these were acknowledged or addressed satisfactorily:

> *As soon as I send a message it is answered instantly . . . so he's been spot on really in terms of kind of connecting in that regard (FGD B)*

Some reported changes being made as a result of their feedback, and felt that these had taken place within an acceptable time-frame. Others reported wanting instant answers to their queries, particularly when making contact on behalf of the audience with whom they would share information. This was especially pertinent in cases where the participant was a point of contact for adults living with long-term conditions or disability and trying to safeguard their wellbeing while navigating COVID-19 guidance:

> *. . .when you think somebody in a stressful situation at home feeling at risk. . .you wish you could just get an answer for them straight away, you know (Interview3)*

Frustration with what were perceived as vague or delayed responses was reported by some participants:

*The feedback loop, I found actually wasn't very effective and you responded back to information that's coming out to ask for either clarification on a few things, and it was delayed by quite a considerable amount of time or not there at all (FGD A)*

Some expressed discontent as they felt that communication between the organisers and the volunteers could be improved. Expectations varied in this regard–some stated that email responses within three working days was acceptable, where others hoped they would get immediate replies.

A few volunteers described using their own resources at times, including use of their own networks or information sources to seek answers or access to the volunteer programme, generating their own dissemination material, and using personal funds:

*Out of my own pocket, I had to buy printer cartridges. . . I am not moaning about the cost-. . .The only way I could get it to the people was if I printed it off. . . it would have been nice if they had said OK, here is the money for the costs or if they had send me some leaflets (Interview2)*

In accordance with COVID-19 guidance, interaction between the volunteer programme and the volunteers was entirely remote, using only emails and webinars. One participant described difficulty with this, namely that they and their respective audience lacked reliable internet access. This affected the level of support this participant was able to access, impacting upon their general satisfaction. Signposting to online resources became a source of frustration and other forms of contact, such as telephone calls, were preferred:

*Birmingham Council need to understand that I do not have regular access to the internet and that I have limited access to emails. Also as much as I want to take part in conference calling I do not have a web cam. (Interview2)*

In terms of volunteer oversight by the programme organisers, participants' preferences varied. Many were satisfied with the degree of freedom they had as a volunteer, particularly as the way the volunteer role was performed varied greatly from person to person:

*I don't think that the they proactively offer support if I'm honest and but I can't say I've needed it either (FGD A)*

*It would have been nice to have had a phone-call to. . . discuss what successes you've had and what should be worked on. . . but some people would say, hang on a minute, I am a volunteer, I'm not an employee of the council (Interview2)*

For example, some participants described simply forwarding the emails they had received from the Birmingham City Council Public Health Team while others would select information and share with their networks either face to face or in their remote social spaces such as zoom calls or group chats. Participants described how their audience ranged from their immediate personal and social networks to those reaching large numbers of people through their professional connections. Some reported that greater efforts to coordinate volunteer activity would have been appreciated and the absence of this implied they were not a priority:

*I felt like they had other priorities. . . we were not top of the list (Interview2)*

Participants were also concerned that nobody had checked what they had been doing in their roles or monitoring their impact:

*No one's ever come and asked me actually saying, well what area do you actually cover you know what domain. . . but no one's ever asked those questions (FGD A)*

The use of peer support was discussed, and some participants felt that the opportunity to pool volunteers' skills, resources, and efforts had been missed, especially in terms of accessing other languages or teaming up to maximise impact where multiple volunteers lived in the same area. Some felt a coffee morning would have allowed volunteers to network, share information, and boost morale. One participant reported that they had agreed to share their contact information for this purpose, but this had not been used:

*There was a suggestion I think from the public health team that a WhatsApp group was set up, and I think the intention behind that was for some sort of peer support so that we could chat and swap experiences and things and I don't know what the outcome of that (FGD A)*

Others felt that peer support would require greater time commitment than they could spare, that they prioritised access to programme organisers instead, and were unsure that this would have been of assistance:

*Peer support, but again, it means extra time on our part to have that commitment to wade through all these WhatsApp messages in the hope that they would be helpful (FGD A)*

*I don't think that that would we would feel satisfied with that, . . . it just feels like peer support might be a bit like the blind leading the blind (FGD A)*

Some acknowledged that others may however find it helpful.

## Programme organisation

The programme consisted of regular and frequent distribution of the latest COVID-19 guidance, shared via email and webinars. Some participants remained satisfied with the programme organisation and did not note any changes, while others reported how the regularity, frequency, and content of these had changed over time:

*We don't get the quick updates. . . we're not getting any of that anymore, we get the Friday "Dear John" email. Really it looks the same it's not even different. . . looks boring (FGD C)*

The content was seen as timely, informative, with local data and a great degree of detail that allowed volunteers to engage to the extent that they needed for their particular purposes. The webinars and emails were regular and predictable. For some, over time these became less predictable, less detailed and less locally relevant, and were shared in formats that were less conducive to sharing with their networks:

*They used to send a link to the City Council website and you could actually save it as a PDF which in some cases, is quite easy, which is convenient to send to other people (FGD B)*

Some participants felt the information they received had become similar to that which would be available from ordinary sources such as media outlets, or of a lesser quality. Webinars ceased to be recorded and topics unrelated to COVID-19 were being included:

*The email that came out last night for argument's sake, there is possibly. . . Oh, I would say, probably a third of the email was COVID-related. . . it's [like] they use it almost as a document actually to sort of advertise other elements that's going on within Birmingham (FGD A)*

This represented a change to what participants had come to expect during their time on the programme or the extent to which they were able to engage, and had implications for their personal investment in the programme. Some participants felt the programme organisers were failing to keep up their side, as volunteers would be happy to "do their bit" if the organisers did the same. Some participants interpreted this as an indication the programme was soon to finish:

*What I'm feeling at the moment when I made the point about the messages are getting watered down and it's almost as though actually we can stop doing this now, (FGD A)*

## Feeling valued

Several participants reported having had a positive experience of being a volunteer, that they had been impressed with the programme, and that they had and would recommend it to others:

*I have recommended already to a few friends and I think one of them joined (Interview 3)*

For some, the changes in the programme over time and that the role would be no longer needed meant they would not encourage others to join:

*I wouldn't at this stage. . . (FGD C)*

Participants reported varying responses from their target audience and those close to them, some had experienced positive feedback and people being warm and receptive, whereas others had experienced hostility, scepticism, or indifference:

*I think I'm lucky that I get such good feedback from people, because it really you know, makes me warm inside. . . I've had lots of different thank you messages. (Interview3)*

*Some people were laughing and . . .making fun of me. . . they were sending me like threats around social media (FGD C)*

Those receiving positive feedback had been sharing information going door to door, or when people were aware of the individual being a Community Champion and would seek information from them and voice their appreciation. Some were unsure of their impact while others had received a lot of appreciation from their target group as they had felt forgotten during the course of the pandemic. Several described how they did not need affirmation or validation for their volunteering role from the programme organisers or others, and that they were not motivated by this but were driven by the desire to help others:

*I don't need a pat on the back or a letter of authority or anything (Interview1)*

Participants reported receiving recognition for their role from programme organisers during webinars and through sharing of encouraging emails. Some felt the programme organisers had shown their appreciation for the participants' work by the time and care they had taken to plan the programme, share the information, run the webinars, and by commissioning an independent programme evaluation:

*Well they are very encouraging, with their emails and their webinars (FGD3)*

Participants were keen to share their experiences with the Birmingham City Council Public Health Team and their positive impact and the receipt of heartening feedback from their target audiences. One participant had shared their experience with the programme organisers and was pleased that their contribution had been included in the programme newsletter. Others described feeling disheartened following failed interactions with programme organisers, for example being unable to contribute during webinars due to attendance numbers and receiving no response when sharing the thank you messages received from the groups with whom they had worked. Some felt that recognition from the programme organisers was lacking, that the thanks received to date was impersonal and not proportionate to the level of work involved and commitment made by the volunteer:

*We need is the care back, feel wanted, the cuddling type, things like that, to say we're so proud of you to come along and things like that, not just here's the information (FGD C)*

For some of those who felt that appreciation had been shown through the careful organisation of the programme, the gradual changes in the programme over time and introduction of new topics was interpreted as a watering down of the role.

## Continued need for the role

Focus groups and interviews had taken place from July-September 2021, at a point at which the COVID-19 pandemic was in a state of change. In England, government-imposed restrictions were being eased and participants reflected on this and their continued efforts as a volunteer. Participants felt that there would be ongoing interest from the programme organisers in maintaining access to a pool of volunteers. However, several expressed concern about the ongoing pandemic and felt there was still a need for their role to remain specific to COVID-19. With the increasing inclusion of non-COVID-19 information in the emails and webinars, some participants believed the focus of the programme had changed, acknowledging that the city was opening up and there was less of the initial urgency observed compared to when the programme started:

*There is a bit of an uncomfortable shift there that we're saying we need this information and they're starting to bring in other things (FGD A)*

*There's a there's a relaxation at the moment, whereas there was an urgency in the initial stages, because not a lot of people knew anything about this. . . The foot's been taken off the accelerator a little bit (FGD A)*

Some participants explained how the basis for their participation was COVID-19 related activity, and reported that the change in focus caused them to question their continued involvement and the value they were adding. Others were worried the information they shared in their volunteer capacity would become less relevant to their audience with the increased inclusion of new, unrelated topics:

*If I start to fill up their inbox is actually with non-specific you know COVID information, then I think you're watering the message, down to the degree, where people might start you know not reading what I'm sending out (FGD A)*

*I signed up to be a COVID champion and so I'm not terribly comfortable now that a lot of the information is about other things. . . I can understand that they see us as a natural conduit to do it, but personally that's not what I signed up for (FGD A)*

Some discussed their interest in expanding their role to other COVID-19 efforts:

*. . .Assist with being a mystery "shopper" to ensure businesses are complying with safe practices i.e, going out to check whether sanitisers are being refilled, premises are being kept clean, beauty salons are being cleaned after each customer, encouraging ventilation in restaurants/ hotels/trampolining sites etc (Interview5)*

Participants discussed how their role might evolve to reflect a new stage in the pandemic where the population would have to adjust to COVID-19 being part of daily life or the implications of Long-COVID. This role evolution had taken place in earlier phases of the pandemic in order to reflect changes to official information and guidance such as initial guidance around Personal Protective Equipment (PPE), then testing, then vaccination. Some described how the groups with whom they shared information were starting to move on and forget, and that the volunteers still had a role to remind people or assist in navigating the changing government guidance.

Many stated that their permission would be required if their role were to move to a non-COVID-19 focus:

*. . .use a separate invite the people who are COVID champions to say, would you be comfortable in also being a Commonwealth Games champion as well (FGD A)*

In the case where non-COVID-19 topics were introduced without the volunteers' permission, this caused resentment for some participants. They reported frustration with the inclusion of new topics and worried that this meant that the programme in its original COVID-19 related form was nearing its end.

### Interest in new responsibilities

When asked about interest in topics unrelated to COVID-19, a number of participants reported that they already advocated or shared information about non-COVID-19 public health issues and/or were involved in broader activities outside of their role as a COVID-19 volunteer:

*I'll put stuff around people having smear tests. . .I mean for years and years and years I was always promoting wellbeing stuff so back in the day when I was in the offices (Interview3)*

These individuals were keen to hear about new opportunities and felt that sharing this information would be manageable, formally within their volunteer role or otherwise:

*We have got the community's attention, so things like smoking, vaping, heart issues. . . this is an opportunity for BCC to take it to the next level.. not just about COVID but health issues in general (Interview2)*

When discussing interest in advocating on behalf of their communities, some participants were wary, stating that individuals in local government were better placed and had a mandate to do this:

*In theory, yes, but I think that when you've got that there's got to be some accountability as to you know if I if I start spouting on about this. . . what's my mandate for doing that (FGD B)*

When discussing interest in sharing other local information unrelated to COVID-19, they felt that existing neighbourhood groups and local news fulfilled this function:

*. . . we already do that through other means, particularly through West Midlands police for argument's sake and through our local community police officer (FGD A)*

All participants stated that inclusion of new public health topics and the addition of new activities to their roles would require their permission. Ideally, this would be done through a process of opportunities being offered and the volunteer being able to consider their interest in their own time:

*. . .whatever I do will need to fit in with the demands on my time that I have already really. (Interview1)*

Finally, participants described how if they were to formally expand their volunteer role, this would require more support from the programme organisers. One participant stated that in this case, the programme would benefit from an expenses reimbursement policy and another that the programme could explore buying out volunteers' time from their professional role:

*It is voluntary. . .but a method of paying expenses and a method of providing resources [would be good] (Interview2)*

## Discussion

Our findings highlight a breadth of experiences when volunteering on this programme and have identified a range of factors relating to volunteer recruitment, retention, and re-purposing, including promoting their engagement with the programme and the topics of interest and minimising demotivation. The period in which the COVID-19 pandemic has taken place is a unique point in history. The pandemic changed volunteering behaviours in the UK, formal volunteering fell to its lowest while more people informally volunteered. However, interest in certain formal volunteering campaigns such as the Birmingham COVID-19 Community Champion programme and the NHS Volunteer Responder scheme withstood this trend because of the spontaneous emergence of those keen to join efforts related to the pandemic. There are numerous examples where interest in volunteering has peaked in relation to a specific focus such as sporting events or natural disasters, from which the resulting interest in volunteering may be catalysed and directed to other issues. While some of the learning from this research is specific to the volunteer programme in question and the context of the COVID-19 pandemic, there are lessons that can be extended to other scenarios and settings.

To retain existing volunteers, maintaining the terms of engagement upon which they joined the programme is important. In this case, the programme stated that it would interact through emails and webinars, so regulating email and webinar frequency, availability, and content, and monitoring and responding to emails from volunteers may encourage their continued engagement on those original terms. If there are changes to how interactions will take place or the content of the programme, this needs to be done through a consultation process, allowing volunteers the opportunity to "opt-in". Among the participants involved, there were those who were particularly keen or well-positioned to undertake new activities. An opt-in process would

allow explicit identification of these individuals and discussion of how their role may change. In doing so, those who are not interested in new activities are not alienated, potentially undermining their undertaking the original brief. Supplementing the original terms to ensure there is a mechanism so that volunteers do not experience out of pocket expenditure may promote long-term engagement and demonstrate appreciation for their input [28]. Use of personal funds for the fulfilment of volunteering roles may undermine their sustainability. Organising events to foster connections between volunteers and boost morale or gestures of thanks and appreciation were important to our participants and may further promote retention [29]. Compared to paid employees, the terms with which volunteers engage with the organisation consist of the obligations, rights and rewards that are believed to be owed in return for their effort [30]. This was seen in our research when participants described how frequently and regularly sharing information remaining relevant to COVID-19 and without other content was a representation of the organisation upholding what had been originally agreed, and in doing so they would be happy to continue undertaking their volunteer role.

Drawing on exploring volunteer retention in this case study, it may be that when recruiting new volunteers, clarifying the role and the terms of engagement from the outset will ensure that volunteers are aware of what is expected of them and what to expect. In our case-study, stating the COVID-19 specific or general public health focus is important, so potential volunteers can assess alignment with their own interests and whether they wish to proceed. This research indicates that, whilst necessary in the circumstances, the programme was limited by its online-only approach, limiting the extent to which volunteers and their audience could engage if they had limited access to the internet. Use of accessible and shareable formats and different languages would appeal to different volunteers and increase reach to previously inaccessible groups. Investment and a specific strategy to target particular marginalised or excluded groups can foster collaboration that may maximise impact of interventions [31]. When considering how to expand the role of volunteers on ongoing programmes, working with communities and possibly other existing volunteer organisations or reflecting on how activities would complement, may be useful [32].

To re-purpose volunteers already on a programme or widen their role, this should be done through a period of consultation, undertaking changes to their remit with their permission. This can be done through offering potential opportunities to allow individuals to decide if these align with their existing interests, promoting longevity in the role. Changes in role should be made following an assessment of how support structures should be adjusted. This might include a payment or reimbursement structure, a re-evaluation of the potential non-monetary benefits for volunteers taking on additional responsibilities, changes in oversight, management, and coordination and uses of different forms of engagement. In our case-study, the non-monetary benefit to volunteers and a key recruitment driver was the immediate access to up-to-date high-quality and locally relevant information. Participants described their dismay at changes in their access to this and their subsequent de-motivation. As such, consideration of this for both retention, recruitment, and re-purposing volunteers is very important [28, 33]. Changes in support structures should be commensurate with changes of role, particularly because appropriate support and management has been cited as a means to promote retention during the volunteers' journey [18]. Our participants discussed the importance of a vetting process when graduating volunteers' responsibilities to a position of advocating on behalf of their community, and checking their credentials and mandate. In particular, it may be important to ensure the new volunteer position is clarified in relation to other similar, possibly paid positions, within the community. While managing relationships between paid and volunteer staff within the same organisation has been extensively studied [34, 35], interactions between volunteers and those in paid positions out in the community are yet to be explored.

PPI greatly strengthened the conduct of this work. Meeting regularly throughout the course of the research enabled input at every stage despite it being a rapid evaluation. The PPI group were able to draw on their own experience of being volunteers and share insights into how to minimise social desirability bias and promote honest sharing during the discussions. This study used rigorous methodology and drew on experiences from a wide breadth of participants. The focus group methodology uses a group interview process that explicitly includes and uses the dynamism of group interaction to generate data [36]. This means that participants are able to explore and clarify their own views and issues of interest to them, using their own vocabulary [37] while the researcher takes a facilitative role [38]. The aim was to represent the widest range of perspectives and experiences [39] and efforts were made to enable participation from a diverse range of participants. Due to the rapidly evolving COVID-19 context and the need for findings and recommendations to be quickly implemented ahead of outdating, rapid qualitative methods were adopted. These have been shown to generate comparable findings to in-depth methods [25, 40]. However this has some implications for the length of time permitted for data collection and how data saturation was interpreted. The data collected from participants at the start of the data collection period started to differ to those collected towards the end due to the changing COVID-19 context. As such, in addition to observing that there was no further emergence of new themes, data collection was ended to ensure a degree of commonality of experience and generalisability of findings across the sample. While our participants represented a broad range of demographic characteristics, the extent to which they represented the wider cohort of volunteers on the programme is unclear as these data are not known. As a result, whether the pool of volunteers are typical of the wider Birmingham population is a possible avenue for future research. It may be that an online-based volunteer programme may be accessible to only specific sections of the local population. Though efforts were made such as offering interviews and discussions at a range of times to accommodate the schedules of participants, and one-to-one and email interviews were offered those unable to attend a focus group, these were all within the hours of 09.00 to 19.00, those working office hours or with caring responsibilities may have been unable to participate. This also meant a range of different qualitative data was collected and analysed together, which must be considered. A further limitation is that the study is at risk of self-selection and social desirability bias, whereby participation was driven by a pre-existing interest in this work or portrayal of positive behaviours due to the nature of the study. However, the wide divergence of views indicates that this may not have been the case. Additionally, the transferability of these findings beyond the COVID-19 context is evidenced by their resonance with the current literature [28–35]. However, the applicability of these findings to other cultures would need to be interrogated as the programme took place in and participants were recruited from the UK, a high-income country.

## Conclusions

Our aim was to explore the views and experiences of individuals involved in public health volunteering during the COVID-19 pandemic and identify factors relating to retention, recruitment, and expansion of their role. Our findings indicate that maintaining the original terms of engaging with volunteers, adjusting these through consultative processes with an emphasis on seeking permission, and ensuring a reliable and consistent management and support structure are conducive to volunteer retention, recruitment, and re-purposing.

## Supporting information

**S1 Appendix. Topic guide.**
(DOCX)

**S1 Table. COREQ (COnsolidated criteria for REporting Qualitative research) checklist COREQ checklist.**
(DOCX)

## Acknowledgments

We would like to thank the National Institute for Health Research ARCWM Public Health Theme Patient and Public Involvement group for their invaluable contributions. The views expressed in this article are those of the author(s) and not necessarily those of the University of Birmingham, National Health Service (NHS), National Institute for Health Research (NIHR) or the Department of Health and Social Care.

## Author Contributions

**Conceptualization:** Ameeta Retzer, Habib Ullah, Modupe Omonijo, Justin Varney, Kate Jolly.

**Data curation:** Ameeta Retzer, Janet Jones, Habib Ullah.

**Formal analysis:** Ameeta Retzer, Janet Jones, Kate Jolly.

**Methodology:** Ameeta Retzer, Sarah Damery, Modupe Omonijo, Justin Varney, Kate Jolly.

**Project administration:** Ameeta Retzer.

**Supervision:** Justin Varney, Kate Jolly.

**Validation:** Sarah Damery.

**Writing – original draft:** Ameeta Retzer.

**Writing – review & editing:** Janet Jones, Sarah Damery, Habib Ullah, Modupe Omonijo, Justin Varney, Kate Jolly.

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
