## [Decision Letter · Decision Letter 0]

5 Jan 2022

PONE-D-21-37424Retaining Public Health Volunteers beyond COVID-19PLOS ONE

Dear Dr. Jolly,

Thank you for submitting your manuscript to PLOS ONE. After careful consideration, we feel that it has merit but does not fully meet PLOS ONE’s publication criteria as it currently stands. Therefore, we invite you to submit a revised version of the manuscript that addresses the points raised during the review process.

We look forward to receiving your revised manuscript.

Kind regards,

Michio Murakami

Academic Editor

PLOS ONE

Journal Requirements:

2. Please provide additional details regarding participant consent. In the ethics statement in the Methods and online submission information, please ensure that you have specified whether: 1) whether the ethics committee approved the verbal/oral consent procedure, 2) why written consent could not be obtained, and 3) how verbal/oral consent was recorded. If your study included minors, please state whether you obtained consent from parents or guardians in these cases. If the need for consent was waived by the ethics committee, please include this information.

[We would like to thank the National Institute for Health Research ARCWM Public Health Theme Patient and Public Involvement group for their invaluable contributions. This study was funded by the National Institute for Health Research (NIHR) Applied Health Research Collaboration West Midlands. The views expressed in this article are those of the author(s) and not necessarily those of the University of Birmingham, Nation Health Service (NHS), National Institute for Health Research (NIHR) or the Department of Health and Social Care.]

 [This study was funded by the National Institute for Health Research (NIHR) Applied Health Research Collaboration West Midlands.]

Additional Editor Comments:

I recommend that the authors summarize the findings in one figure or table in the main body of the paper; the Appendix is well organized, but it would be preferable to include the concise figure (or table) in the main body of the paper so that the reader can grasp the findings of this paper at a glance.

Please explain what the authors mean "FGD A" etc. in the Appendix.

Reviewers' comments:

Reviewer's Responses to Questions

**Comments to the Author**

1. Is the manuscript technically sound, and do the data support the conclusions?

Reviewer #1: Yes

Reviewer #2: Yes

2. Has the statistical analysis been performed appropriately and rigorously? 

Reviewer #1: N/A

Reviewer #2: Yes

3. Have the authors made all data underlying the findings in their manuscript fully available?

Reviewer #1: Yes

Reviewer #2: Yes

4. Is the manuscript presented in an intelligible fashion and written in standard English?

Reviewer #1: Yes

Reviewer #2: Yes

5. Review Comments to the Author

Reviewer #1: Changes in people's volunteering behaviors by the COVID-19 pandemic is an important topic and the manuscript was well written.

However, data collection was made among COVID-19 related volunteers during COVID-19 pandemic and this may make the findings difficult to be generalized, compared to data collection among volunteers engaging other public health issues.

It has been said that the data collection were made until data saturation was reached, however, is there any gender or age specific differences in volunteering motives?

Reviewer #2: I quite enjoyed reading this paper. It serves as an excellent example of informal, spontaneous volunteer engagement for the dissemination of public health information during a crisis Thank you for conducting this independent programme evaluation via a qualitative case study of the COVID-19 Community Champions. Maintaining positive patterns of volunteering for other public health issues beyond the pandemic is novel. Please see attached reviewer's comments for minor edits/suggestions for your revision. Thank you.

6. PLOS authors have the option to publish the peer review history of their article (what does this mean?). If published, this will include your full peer review and any attached files.

Reviewer #1: No

Reviewer #2: **Yes: **Gretchen Roman, PT, DPT, PhD

---

## [Author Response · Author response to Decision Letter 0]

15 Feb 2022

Thank you for the thoughtful and considered feedback. Each of the reviewer's points have been addressed in the attached document.

---

## [Editor Report · Decision Letter 1]

17 Feb 2022

PONE-D-21-37424R1Retaining Public Health Volunteers beyond COVID-19PLOS ONE

Dear Dr. Jolly,

Thank you for submitting your manuscript to PLOS ONE. After careful consideration, we feel that it has merit but does not fully meet PLOS ONE’s publication criteria as it currently stands. Therefore, we invite you to submit a revised version of the manuscript that addresses the points raised during the review process.

In the revised version of the paper, there are responses to the comments from Reviewer 2, but there are no responses to the comments made by Reviewer 1 and the Editor (me).  Please include responses to all the comments.

We look forward to receiving your revised manuscript.

Kind regards,

Michio Murakami

Academic Editor

PLOS ONE

Journal Requirements:

Additional Editor Comments (if provided):

In the revised version of the paper, there are responses to the comments from Reviewer 2, but there are no responses to the comments made by Reviewer 1 and the Editor (me). Please include responses to all the comments.

---

## [Author Response · Author response to Decision Letter 1]

21 Feb 2022

Thank you for the opportunity to review and resubmit our paper. Please see the attached point-by-point documents that now includes our response to the editorial feedback and comments from reviewer 1.

---

## [Decision Letter · Decision Letter 2]

28 Mar 2022

PONE-D-21-37424R2

Retaining Public Health Volunteers beyond COVID-19

PLOS ONE

Dear Dr. Jolly,

Thank you for submitting your manuscript to PLOS ONE. After careful consideration, we have decided that your manuscript does not meet our criteria for publication and must therefore be rejected.

Specifically:

The statistical analysis specifically approach of the Thematic analysis has not been performed appropriately and interpreted. The issues discussed in the paper are exciting but lacks appropriate analysis and have significant demographic variations in the sample. The interpretations are not in line with addressing the large deviations in the sample. Additionally, the research lacks the presence of any conceptual framework or hypothesis the authors are trying to prove. 

I am sorry that we cannot be more positive on this occasion, but hope that you appreciate the reasons for this decision.

Yours sincerely,

Prabhat Mittal, Ph.D.

Academic Editor

PLOS ONE

Reviewers' comments:

Reviewer's Responses to Questions

**Comments to the Author**

1. If the authors have adequately addressed your comments raised in a previous round of review and you feel that this manuscript is now acceptable for publication, you may indicate that here to bypass the “Comments to the Author” section, enter your conflict of interest statement in the “Confidential to Editor” section, and submit your "Accept" recommendation.

Reviewer #3: All comments have been addressed

2. Is the manuscript technically sound, and do the data support the conclusions?

Reviewer #3: Yes

3. Has the statistical analysis been performed appropriately and rigorously? 

Reviewer #3: N/A

4. Have the authors made all data underlying the findings in their manuscript fully available?

Reviewer #3: Yes

5. Is the manuscript presented in an intelligible fashion and written in standard English?

Reviewer #3: Yes

6. Review Comments to the Author

Reviewer #3: (No Response)

7. PLOS authors have the option to publish the peer review history of their article (what does this mean?). If published, this will include your full peer review and any attached files.

Reviewer #3: No

- - - - -

---

## [Author Response · Author response to Decision Letter 2]

5 Jul 2022

Dear Editor,

Thank you for the opportunity to engage in the peer review process in relation to this

paper, “Retaining Public Health Volunteers beyond COVID-19” (PONE-D-21-

37424R2). Both peer reviewers stated that our manuscript was well-written, focused

on an important topic, and served as “an excellent example of informal, spontaneous

volunteer engagement for the dissemination of public health information during a

crisis”. Peer reviewer feedback requested the addition of some detail about the

COVID champions volunteer programme, and some minor clarifications to the text.

Neither reviewers raised any concerns about the methodological conduct of the

study, stating that the manuscript was technically sound and the data supported the

conclusions. The previous editor requested minor clarifications, which we provided.

Therefore, we were extremely disappointed to receive the most feedback from a new

editor, and the decision to not proceed with publication. We believe the issues raised

strongly contradict the judgment of the manuscript received to date from the peer

reviewers and previous editor, thus we wish to formally appeal the decision. We

present a point-by-point response to the recent comments below.

Editorial comment: The statistical analysis specifically approach of the

Thematic analysis has not been performed appropriately and interpreted.

Thematic analysis is a form of qualitative study and is not a quantitative, statistical

approach that aims for generalisability. It was undertaken in this research by two

experienced qualitative researchers and reported transparently in accordance with

the consolidated criteria for reporting qualitative research (COREQ) checklist. The

completed checklist has been submitted along with the manuscript as an appendix.

The ‘rapid’ qualitative method reported in this manuscript pertained only to the use of

field-notes rather than verbatim transcripts, alongside a pragmatic approach to

participant recruitment, both of which were responses to the rapidly evolving context

of the pandemic. This was to ensure the experiences captured from participants

were comparable and the findings were not outdated. Rapid qualitative methods

have been found to be methodologically sound, all data were analysed and

interpreted following standard and accepted methods of thematic analysis, and

our approach is outlined in detail in the manuscript.

Editorial comment: The issues discussed in the paper are exciting but lacks

appropriate analysis and have significant demographic variations in the 

sample. The interpretations are not in line with addressing the large deviations

in the sample.

Sample size calculations are not used in qualitative research, instead recruitment

and sample size are guided by the concept of data saturation. This is widely used in

qualitative research and refers to the point at which no new information is discovered

in data analysis. This is reached through a deliberate effort to recruit participants

across a wide range of demographic characteristics. This is associated with the

premise of qualitative research being to capture and understand the breadth and

diversity of experience rather than to quantify those experiences in a statistical

sense. The interpretations reported are directly from these data and a robust

approach was used through mechanisms such as double-coding, as per accepted

qualitative practice.

Editorial comment: Additionally, the research lacks the presence of any

conceptual framework or hypothesis the authors are trying to prove.

The lack of a conceptual framework/hypothesis is because the approach used was

largely inductive rather than deductive, i.e. we allowed the data to determine the

themes as we did not have pre-conceived themes we expected to find. However, we

cite the use of the Gaskin model of volunteer involvement to inform our

understanding of the topic and to formulate the initial coding frame in addition to the

research aims. A flexible iterative approach was used to allow for the exploration

of novel themes. This is an accepted and widely used approach to explore topics

with limited evidence base and diversity of experiences.

We respectfully request that the decision to reject this manuscript is reconsidered in

light of the information we have provided. We believe this topic is of interest, the

methods used are robust and have been reported transparently.

With thanks and best wishes,

Ameeta

---

## [Decision Letter · Decision Letter 3]

19 Apr 2023

PONE-D-21-37424R3

Retaining Public Health Volunteers beyond COVID-19

PLOS ONE

Dear Dr.Jolly,

Thank you for submitting your manuscript to PLOS ONE. After careful consideration, we feel that it has merit but does not fully meet PLOS ONE’s publication criteria as it currently stands. Therefore, we invite you to submit a revised version of the manuscript that addresses the points raised during the review process.

I have no significant concern relating to your revised manuscript,but,on respect and transparency for any reviewer's comments and contribution,I kindly invite you to refer to the comments of Reviewer 4 and 5 for a paragraph clarification regarding their concerns.

Thank you for choosing PLOS to submit your valuable work!

We look forward to receiving your revised manuscript.

Kind regards,

Silva Ibrahimi, PhD

Academic Editor

PLOS ONE

Journal Requirements:

1. 1. We notice that your manuscript file was uploaded on Feb. 21, 2022. 
Please can you upload the latest version of your revised manuscript as the main article file, ensuring that does not contain any tracked changes or highlighting. This will be used in the production process if your manuscript is accepted. Please follow this link for more information: http://blogs.PLOS.org/everyone/2011/05/10/how-to-submit-your-revised-manuscript/

Additional Editor Comments (if provided):

Reviewers' comments:

Reviewer's Responses to Questions

**Comments to the Author**

1. If the authors have adequately addressed your comments raised in a previous round of review and you feel that this manuscript is now acceptable for publication, you may indicate that here to bypass the “Comments to the Author” section, enter your conflict of interest statement in the “Confidential to Editor” section, and submit your "Accept" recommendation.

Reviewer #4: (No Response)

Reviewer #5: (No Response)

Reviewer #6: (No Response)

2. Is the manuscript technically sound, and do the data support the conclusions?

Reviewer #4: Yes

Reviewer #5: Partly

Reviewer #6: Yes

3. Has the statistical analysis been performed appropriately and rigorously? 

Reviewer #4: Yes

Reviewer #5: N/A

Reviewer #6: N/A

4. Have the authors made all data underlying the findings in their manuscript fully available?

Reviewer #4: Yes

Reviewer #5: No

Reviewer #6: Yes

5. Is the manuscript presented in an intelligible fashion and written in standard English?

Reviewer #4: Yes

Reviewer #5: Yes

Reviewer #6: Yes

6. Review Comments to the Author

Reviewer #4: Explain in the methodology part how many participants involved in semi-structured focus group discussions, one-to-one interviews and email interviews

Reviewer #5: Dear author

Your article addresses an important issue regarding volunteer work.

Some observations and suggestions to your article:

1. Qualitative data collection was carried out using different methods, but they were analyzed and computed in the same way. The collection from a focus group brings a very different result from the collection by individual interview. In the focus group, the opinions expressed are influenced by the interaction of the participants, so I would not recommend the use of different methods for this issue.

2. I would include in the methods which were the guiding topics of the focus group and which were the questions of the individual interviews.

3. I would be cautious in generalizing the results to other cultures.

Reviewer #6: This is a well-written manuscript.

The study objective is clear and the methods used are appropriate and well detailed.

In my opinion, the revisions made to issues made by previous reviewers are sufficient.

I recommend for acceptance.

7. PLOS authors have the option to publish the peer review history of their article (what does this mean?). If published, this will include your full peer review and any attached files.

Reviewer #4: No

Reviewer #5: No

Reviewer #6: No

---

## [Author Response · Author response to Decision Letter 3]

29 Sep 2023

Dear Editor,

Thank you for the opportunity to respond and address these comments. Please see our responses below:

Reviewer #4: 

Explain in the methodology part how many participants involved in semi-structured focus group discussions, one-to-one interviews and email interviews

Author Response:

Thank you – this has been reported in the results section, broken down by focus group discussions, interviews, and email interview.

Reviewer #5: 

Your article addresses an important issue regarding volunteer work.

Some observations and suggestions to your article:

1. Qualitative data collection was carried out using different methods, but they were analyzed and computed in the same way. The collection from a focus group brings a very different result from the collection by individual interview. In the focus group, the opinions expressed are influenced by the interaction of the participants, so I would not recommend the use of different methods for this issue.

2. I would include in the methods which were the guiding topics of the focus group and which were the questions of the individual interviews.

3. I would be cautious in generalizing the results to other cultures.

Author Response:

Thank you for this feedback. 

1. The decision to undertake both interviews and focus group discussions was a pragmatic one intended to accommodate as many people as possible in the rapid evaluation. We have included the combination of the two forms of data as a limitation in our discussion (p15) “This also meant a range of different qualitative data was collected and analysed together, which must be considered.”

2. The topic guide has now been added as an appendix and a statement has been added to the methods section to clarify that the same topic guide was used for each (p5) “The same topic guide was used for the focus groups, one-to-one interviews and email interviews (Appendix 1).”

3. Thank you – this has now been explicitly stated in the discussion (p15) “However, the applicability of these findings to other cultures would need to be interrogated as the programme took place in and participants were recruited from the UK, a high-income country.”

Reviewer #6: 

This is a well-written manuscript.

The study objective is clear and the methods used are appropriate and well detailed.

In my opinion, the revisions made to issues made by previous reviewers are sufficient.

I recommend for acceptance.

Author Response:

Thank you for this feedback. 

With thanks and best wishes,

Ameeta

---

## [Decision Letter · Decision Letter 4]

27 Oct 2023

Retaining Public Health Volunteers beyond COVID-19

PONE-D-21-37424R4

Dear Dr. Jolly

We’re pleased to inform you that your manuscript has been judged scientifically suitable for publication and will be formally accepted for publication once it meets all outstanding technical requirements.

Kind regards,

Silva Ibrahimi, PhD

Academic Editor

PLOS ONE

Additional Editor Comments (optional):

Reviewers' comments:

Reviewer's Responses to Questions

**Comments to the Author**

1. If the authors have adequately addressed your comments raised in a previous round of review and you feel that this manuscript is now acceptable for publication, you may indicate that here to bypass the “Comments to the Author” section, enter your conflict of interest statement in the “Confidential to Editor” section, and submit your "Accept" recommendation.

Reviewer #4: All comments have been addressed

2. Is the manuscript technically sound, and do the data support the conclusions?

Reviewer #4: Yes

3. Has the statistical analysis been performed appropriately and rigorously? 

Reviewer #4: Yes

4. Have the authors made all data underlying the findings in their manuscript fully available?

Reviewer #4: Yes

5. Is the manuscript presented in an intelligible fashion and written in standard English?

Reviewer #4: Yes

6. Review Comments to the Author

Reviewer #4: The author manages to answer all the comments given by the reviewers. This study can benefit the society especially in the voluntary program.

7. PLOS authors have the option to publish the peer review history of their article (what does this mean?). If published, this will include your full peer review and any attached files.

Reviewer #4: No

---

## [Editor Report · Acceptance letter]

2 Nov 2023

PONE-D-21-37424R4 

Retaining Public Health Volunteers beyond COVID-19 

Dear Dr. Jolly:

I'm pleased to inform you that your manuscript has been deemed suitable for publication in PLOS ONE. Congratulations! Your manuscript is now with our production department. 

Kind regards, 

on behalf of

Dr. Silva Ibrahimi 

Academic Editor

PLOS ONE